# Variation and Correlations in Departures from Symmetry of Brain Torque, Humeral Morphology and Handedness in an Archaeological Sample of *Homo sapiens*

**Antoine Balzeau** [1,2,*], **Lou Ball-Albessard** [1,3] **and Anna Maria Kubicka** [1,4]

[1] PaleoFED team, UMR 7194, CNRS, Département Homme et Environnement, Muséum national d'Histoire naturelle, Musée de l'Homme, 17, Place du Trocadéro, 75016 Paris, France; lou.albessard@gmail.com (L.B.-A.); amkkubicka@gmail.com (A.M.K.)

[2] Department of African Zoology, Royal Museum for Central Africa, 3080 Tervuren, Belgium

[3] PalaeoHub, Department of Archaeology, University of York, Wentworth Way, Heslington, York YO10 5DD, UK

[4] Department of Zoology, Poznań University of Life Sciences, Wojska Polskiego 71C, 60-625 Poznań, Poland

* Correspondence: abalzeau@mnhn.fr

**Abstract:** The anatomical asymmetries of the human brain are the subject of a great deal of scientific interest because of their links with handedness and lateralized cognitive functions. Information about lateralization in humans is also available from the post-cranial skeleton, particularly the arm bones, in which differences in size and shape are related to hand/arm preference. Our objective here is to characterize the possible correlations between the endocranial and post-cranial asymmetries of an archaeological sample. This, in turn, will allow us to try to identify and interpret prospective functional traits in the archaeological and fossil records. We observe that directional asymmetry (DA) is present both for some endocranial and humeral traits because of brain lateralization and lateralized behaviors, while patterns of fluctuating asymmetry (FA) vary. The combined study of these anatomical elements and of their asymmetries can shed light on the ways in which the body responds to dependent asymmetrical stimuli across biologically independent anatomical areas. Variations in FA are, in this context, indicators of differences in answers to lateralized factors. Humeri tend to show a much larger range of variation than the endocast. We show that important but complex information may be extracted from the combined study of the endocast and the arms in an archaeological sample of *Homo sapiens*.

**Keywords:** lateralization; directional asymmetry; fluctuating asymmetry; brain endocast; biomechanical properties; human behavior

## 1. Introduction

In prehistoric sciences, the human archeopaleontological record is studied in order to better understand past human societies. The behaviors of prehistoric humans are documented by direct evidence, such as tools, objects loaded with symbolic significance, or cave paintings. The anatomy of humans may also provide important clues, though it is more difficult to interpret [1–4]. Our objective here is to propose an attempt to characterize departures from symmetry for the endocranial and humeral anatomy of an archaeological sample, in order to search for possible identifications and interpretations of prospective functional traits. In this context, we deal with different aspects of asymmetry, looking at potential variations and correlations in directional asymmetries (DA) as indicators of brain/manual

lateralization. We also explore patterns of variations in fluctuating asymmetry (FA) for the different anatomical traits analyzed, in order to discuss variation in the respective biological responses of the brain and arms to asymmetrical stimuli linked to lateralized behaviors. This approach is challenging because we study anatomical areas and traits that are morphologically different from each other, and which require the use of different methodologies. Moreover, these anatomical areas are known to be highly lateralized and subject to high levels of plasticity. From a global anthropological point of view, this study is at the crossroad of various disciplinary fields and, as such, will open important perspectives for future research on the relationship between anatomy and behaviors in past human populations.

The anatomical asymmetries of the human brain have been documented for over a century and are still widely investigated for their functional, physiological and behavioral implications [5]. However, the more specific question of when they appeared in the hominid lineage and in fossil hominins remains controversial [6–10]. An asymmetrical brain could provide an evolutionary advantage through the augmentation of neural capacity, by allowing parallel and separate processing in its two hemispheres [11]. The left hemisphere is specialized for language and speech. It also possesses capacities for solving problems that are crucial for human behaviors. The right hemisphere has its own specializations, performing an essential role in general processes. The two hemispheres are not proficient for the same higher order cognitive processes [12–16] although they are of a roughly similar size and have similar numbers of neurons. Moreover, the connectivity systems of the two hemispheres are different, probably in relation to their respective cognitive functions [17]. In this context, brain size or neuron count alone cannot entirely explain human aptitudes. The study of the brain's structural asymmetries as an anatomical substrate of functional asymmetries in *Homo sapiens*, in comparison with great apes and fossil hominins, is, therefore, crucial to appreciate the structural basis of modern human cognition. The gross asymmetries of brain endocasts (the imprints left by the brain on the internal surface of the skull) are an anatomical feature which has been studied particularly extensively, both on living humans and fossil hominins The combination of right frontal/left occipital protrusions is usually associated to the 'torque' pattern. The larger frontal or occipital projection (petalia) is coupled with another component, the larger width of the more projecting hemisphere relative to the other area (lobar asymmetries). The interest of these anatomical features in the context of paleoneurological studies is that they are easily identifiable on physical or virtual casts of the endocranial surface. The most protruding points of the lobes are always visible and simple to locate. In order to consider the lack of homogeneity in the definition of brain asymmetries and in the methods used to quantify them, we follow here the original anatomical definition of the petalias. Indeed, while LeMay [18,19] considered only the frontal and occipital lobes and their variation in antero-posterior projection, later studies have used the term 'petalias' to refer to bilateral differences in the lateral extension of the posterior area of the frontal lobes [20] or to volumetric variations between the hemispheres [21–25]. It was largely accepted that this pattern of asymmetries appeared with early *Homo* [8,20,26,27] and is more common in right-handed individuals [18,19,28–32]. However, based on an original methodology applied to the largest samples ever used, we have demonstrated a shared pattern of protrusions of the frontal and occipital across all hominids, including modern humans, extant African great apes and fossil hominins [2,33]. These asymmetries are a topic of debate in non-human primate brain studies [18,21,22,32,34–36] and paleoanthropology [6–8,20,26,27,37–39] because of their relationship with handedness and other specific aspects of human cognition. Although there are specific areas of the cortex whose asymmetries are also correlated to handedness, these are not easily delineated on endocasts. This is because the internal cranial surface is separated from the brain by the meninges and cerebro-spinal fluid, leaving relatively faint and variable surface markings on the endocast. Moreover, considering the presence of patterns of gross asymmetry and torque in the endocast, quantifying local asymmetries represents a methodological challenge, and it is not clear what advantage it would provide over the use of the petalias.

Hand preference in the hominin fossil record is particularly difficult to approach. Information about lateralization in hominins is available from the post-cranial skeleton, particularly the arm bones



in which differences in robusticity are related to hand/arm preference. Previous studies have focused on gross external anatomy (see review [40]) or on analyses of single CT slices [1]. A recent in vivo study shows the limitations of earlier methodologies [41] and suggests that 'hand preference' can be successfully inferred from measures of upper limb geometric asymmetry [42]. Relationships between cortical and trabecular architecture and functional patterns are also more complex than previously thought and need further investigation [41,43].

In this context, although some anatomical and structural components of the brain are related to hand preference, they have not yet been precisely delimited and the relationships with details of the surface of the brain endocast remains to be established [2]. The same conclusion can be drawn for the interpretation of hand preference, as deduced from post-cranial elements [42]. The main limitation is, of course, the application of this knowledge, resulting from separate studies (focused respectively on the brain/endocast, on biological skeletal adaptations to lateralized behaviors, and on hand preference) to specimens of past human societies, including fossils from other species than *Homo sapiens*. We propose here an original methodological study looking at departures from bilateral symmetry in different anatomical areas, the endocast and the humeri, in relation with biological and behavioral influences. We will investigate the anatomical asymmetries of the endocast, specifically the petalias because they are known to be related to manual lateralization, as well as the internal parameters of pairs of humeri. In order to do this, we will analyze CT data for an archaeological sample that preserves associated skulls and pairs of humeri. We will, therefore, test if we can identify potential variation and correlations in directional asymmetries for the brain endocasts and humeri which are known indicators of brain/manual lateralization. A second, and larger, objective will be to explore patterns of variations in fluctuating asymmetry for each of the anatomical traits analyzed, in order to discuss possible variation in the respective biological responses of the brain and arms to lateralized behaviors.

## 2. Materials

The analyzed sample consists of 32 adult individuals with a preserved skull and pair of humeri. The population studied was excavated from a cemetery located in Ostrów Lednicki (West Poland) which is an island on Lednica Lake. This archaeological site is dated to a period extending from the 10th to the end of the 13th century [44]. The sex and age of the individuals were assessed according to the standards of Buikstra and Ubelaker [45] and White and Folkens [46] using features of the skull, pelvic morphology and changes in the auricular surfaces of the illium. Only individuals with no observable pathological changes in the analyzed bones (e.g., fractures, osteophytes, porosity) were accepted for this analysis. The individuals from this medieval population belonged to a homogeneous society in terms of their socioeconomic status [47], and their economy was based mainly on agriculture, with the addition of animal husbandry and hunting [48,49].

In order to analyze the association between the asymmetries of the endocast and arms, the skulls and pairs of humeri belonging to 32 adult individuals were scanned using computed tomography (CT). All material was scanned in the same cranio-caudal position with a standard protocol ($0.625 \times 0.625 \times 0.625$ mm$^3$ cubic voxels), using 32-slice CT (Siemens SOMATOM Sensation). Next, for each skull, a reconstruction of the endocast was prepared using the endomaker R package [50,51].

## 3. Methods

Concerning the study of endocranial asymmetries (Figure 1), we used a protocol that we developed to precisely quantify the petalias [2,33]. The objective was to determine the variations in the location of the most protruding points on, respectively, the right and left frontal and occipital lobes (i.e., the frontal and occipital poles, namely the petalias). For that purpose, an external and independent frame of reference was defined based on three anatomical points on the skull (glabella, inion and basion). The pattern of variation of endocranial petalias was detailed by dissociating their antero-posterior, vertical and lateral components. To do so, two lines were constructed virtually between landmarks positioned on the external surface of the skull. The first line (L1) went through the glabella and inion

and the second line (L2) went through the basion and was orthogonal to L1. The most anteriorly protruding points on the left and right frontal lobes and the most posteriorly protruding points on the left and right occipital lobes were projected orthogonally onto L1. The distance on L1 between these bilateral projections corresponds to the antero-posterior components of, respectively, the frontal and the occipital petalia. The four endocranial landmarks were also orthogonally projected on L2. The distance on L2 between these different projections corresponds to the vertical component of the frontal and occipital petalia. Moreover, the four endocranial landmarks were finally projected orthogonally on the plane defined by L1 and L2. The resulting difference between the right and left lateral distances to the plane corresponds to the lateral component of the frontal and occipital petalia. As a result, this protocol allows us to obtain six dimensions to describe the location, in 3D, of the most protruding points on the right and left frontal and occipital lobes. They are the antero-posterior, vertical and lateral components of the frontal and occipital petalias, respectively, labelled as F.AP, F.VERT, F.LAT, O.AP, O.VERT and O.LAT. This protocol has been tested and validated [33], the measurement error was demonstrated to be small, relative to the size of the variations quantified for departures from bilateral symmetry. Additionally, and for comparative purposes with other studies, we have characterized R-L frontal and R-L occipital petalias as qualitatively described by Holloway and de la Coste-Lareymondie [20], taking into consideration the lateral extension of the anterior part of the frontal lobes in superior view.

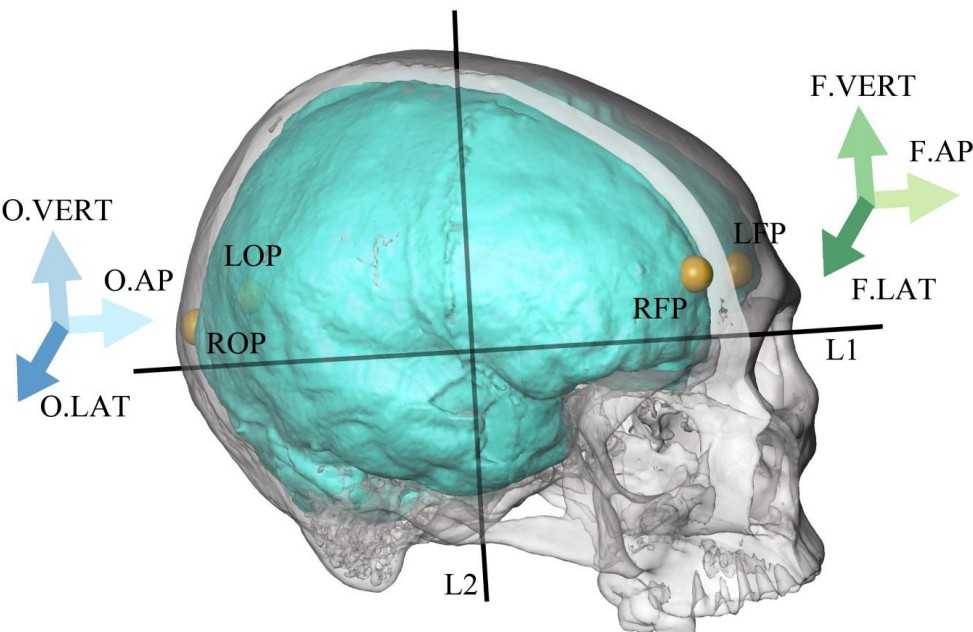

**Figure 1.** Illustration of the protocol designed and used to quantify the brain endocast petalias. Three-dimensional models of the skull (grey) and of the virtual endocast (blue) displayed at different levels of transparency. Right (R), left (L), frontal (F), occipital (O), poles/petalia (P), anteroposterior (AP), vertical (VERT), lateral (LAT). On the skull, a line (L1) is traced through glabella and inion and a second line (L2) is traced through basion orthogonally to the first line. Four landmarks are defined on the endocranial surface (for right (RFP) and left (LFP) frontal poles and right (ROP) and left (LOP) occipital poles). Then, frontal and occipital poles are orthogonally projected on L1. The distance between the projections of the points is defined as the antero-posterior component of the petalia. The four endocranial points are projected orthogonally on L2 and the distance between the projected images of the points corresponds to the vertical component of the petalia. The four points of the brain endocast are projected orthogonally on the plane defined by the two lines and the bilateral difference between the right and left sides for this lateral distance constitutes the lateral component of the petalia. The arrows show the orientation of the different components of the petalias, the color for each arrow is used for each corresponding component in Figures 2 and 3.

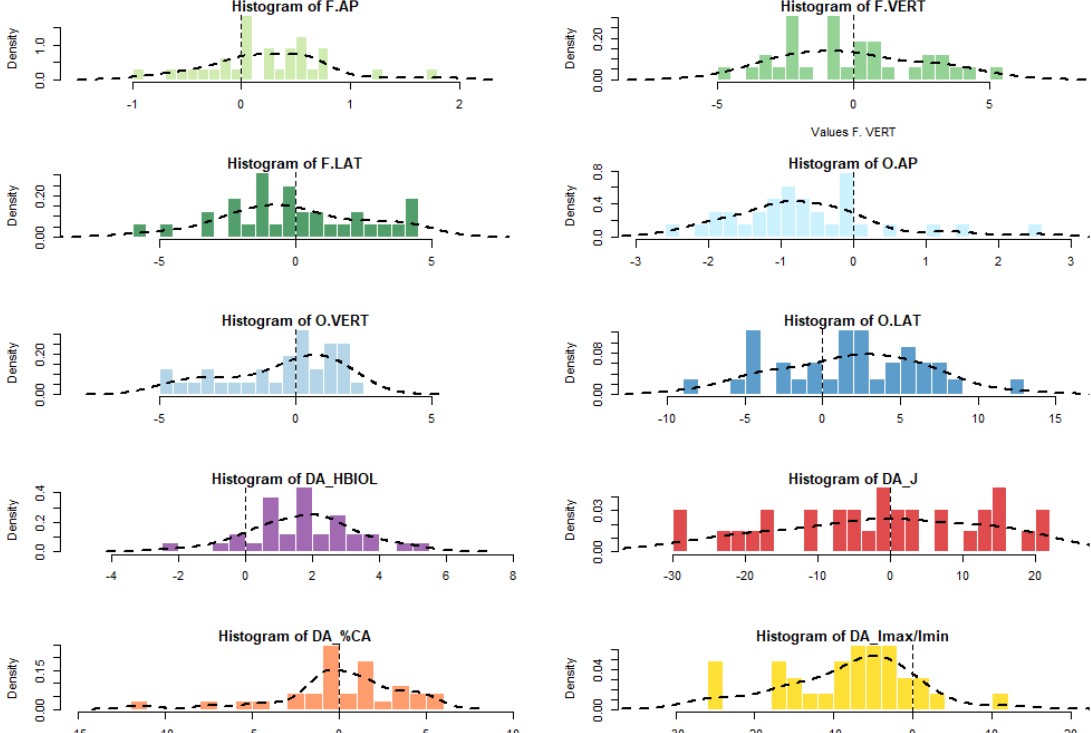

**Figure 2.** Frequency distribution of size-corrected (R-L) components of the petalias (respectively, antero-posterior, vertical and lateral components of the frontal (in different levels of green) and occipital petalias (in different levels of blue) and parameters for the humeri (HBIOL in purple, J in red, %CA in orange and Imax/Imin in yellow)). Frontal (F), occipital (O), anteroposterior (AP), vertical (VERT), lateral (LAT).

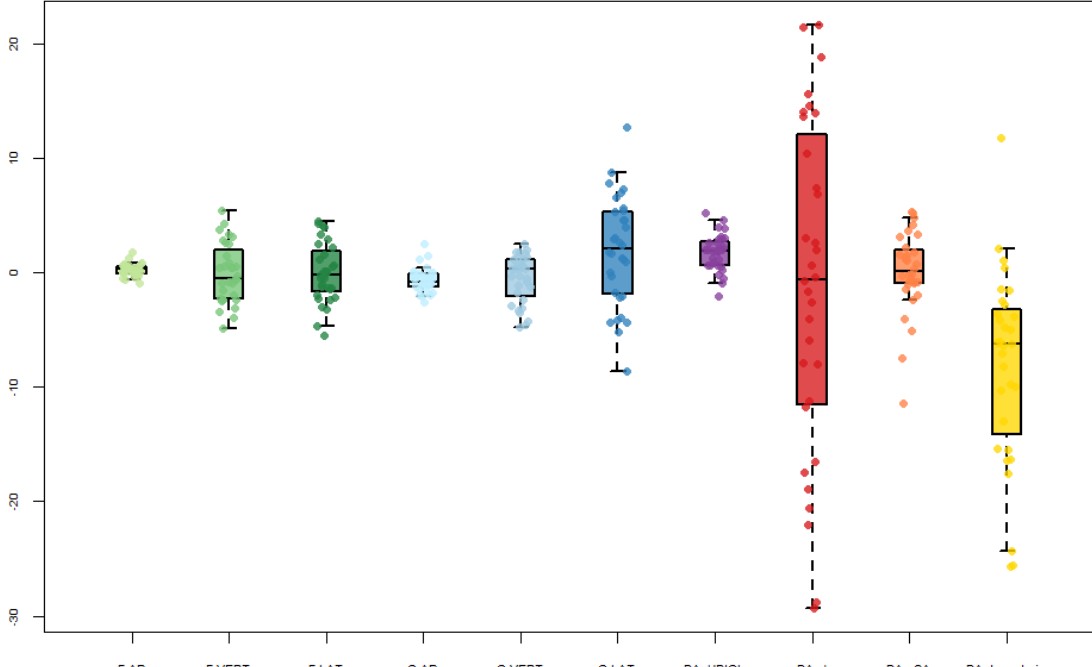

**Figure 3.** Boxplots of size-corrected (R-L) components of the petalias (respectively, antero-posterior, vertical and lateral components of the frontal (in different levels of green) and occipital petalias (in different levels of blue) and parameters for the humeri (HBIOL in purple, J in red, %CA in orange and Imax/Imin in yellow)) showing the median, quartiles, the data points that are most extreme but within the interquartile range, and all data points as individual points (dots).

Concerning the study of the asymmetries between the left and right humeri (Table 1), the cross-section at 35% of the biomechanical length of the bone (HBIOL) was taken using the OsiriX Lite software for each CT scan of the humerus. Following Ruff [52], the length parameter (HBIOL) was measured as the distance between the most proximal point on the humeral head to the most distal point on the lateral lip of the trochlea. Next, the cross-sectional geometric properties were calculated using the NIH Image software with a plugin Moment Macro [53] in order to measure the following variables: the total area (TA), the cortical area (CA), the maximum (Imax) and minimum (Imin) second moments of area and the polar moment of area (J). Furthermore, the following two cross-sectional properties were evaluated: percentage of cortical area (% CA), calculated as %CA = (CA/TA) × 100), and ratio of maximum to minimum second moments of area (Imax/Imin). According to Lieberman et al. [54] and Stock and Shaw [55] these parameters provide the most accurate estimates of cross-sectional geometric properties. The polar moment of area (J) was scaled by powers of biomechanical humeral length before statistical analysis (length raised to the power of 5.33). The upper limb is not weight bearing but some cross-sectional parameters (e.g., J) must be controlled for differences in body size. According to Ruff [52], this method of standardization is less appropriate than controlling properties for body mass, but does not create dimensionless values. Moreover, scaling to bone length provides greater compatibility between the results obtained in this study, and those found in the literature [1,42,52,56]. Ratios of principal moments of area (Imax/Imin) and relative cortical area (%CA) were not standardized either to humeral length or to body mass [42,57].

**Table 1.** Description of the humeral biomechanical properties.

| Abbreviation | Parameter | Description |
|---|---|---|
| HBIOL | the biomechanical length of the humerus | measured from the most proximal point on the head to the most distal point on the lateral lip of the trochlea |
| J | the polar moment of area | indicates resistance to bending and torsional rigidity |
| %CA | percentage of cortical area | calculated as %CA = (CA/TA) × 100, it shows the distribution of cortical area versus total subperiosteal area |
| Imax/Imin | ratio of maximum to minimum second moments of area | calculated as Imax/Imin, ratio close to 1 indicates circular shape of the cross-section |

The anatomical traits quantified on the endocast and the humeri are of different natures. Our protocol for the quantification of endocranial asymmetries has the major advantage of relying on a frame of reference which is not influenced by the global asymmetry of the brain, thus allowing us to quantify petalial components without noise from other gross cerebral asymmetries. Moreover, we quantified, in this study, the difference in location between pairs of significant anatomical landmarks instead of the absolute value of metrical traits on both sides. For this reason, we cannot use some classical descriptors of asymmetries which are calculated using trait size (e.g., see the list in Palmer and Strobeck [58]). In addition, traits quantified on the humeri vary in the amount of developmental instability to a great extent. Fluctuating asymmetry (FA) and, therefore, developmental instability (DI), are influenced by trait size. In this context, we have used conventional FA indices with trait size correction calculated by dividing the individual values by the maximal length of the corresponding skull for the endocast, and by ((R+L)/2) for the humeri traits. This correction is mandatory to allow for a comparison of the degree and extent of variation for the different anatomical traits studied here, and will also allow for future comparisons within the context of new research on FA in humans.

Analyses of FA may be strongly influenced by two sources of error: measurement error (ME) and sampling error [58,59]. Different potential sources of error were tested at every step of the analytical process, which was found to be valid and reproducible (refer to [33], for a detailed description of the methodology for the petalias and to Stock and Shaw, 2007 for the humeri). Other well-known problems [58] exist in the context of FA studies. A single outlier datum has more effect on the variance than on the mean. Grubbs' test statistic [60] was used to recognize the statistically significant outliers.

A non-parametric test of association (Spearman's coefficient of rank correlation) was used to test the relationship between the magnitude of the petalias and those of the traits measured on the humeri. Tests for size dependence were also conducted in order to look at potential correlations between the asymmetries measured on the endocast and the skull but also between asymmetries measured on humeri and trait size before size correction. These tests do not assume homogeneity of variance and, therefore, are likely not influenced by a few extreme observations.

Signed asymmetry is the difference between the right and left sides for each trait in an individual (R-L). This parameter has the advantage to retain information about the direction of the asymmetry. Absolute asymmetry is calculated as the absolute value of the difference between the right and left sides for each analyzed trait in an individual (|R-L|). It also corresponds to FA1 (mean asymmetry) or FA2 (when scaled) but the size correction parameter is not the same for the endocast and the humeri in our study [58]. FA4a, a corrected factor of the variation in mean asymmetry, is calculated using the formula $0.798\sqrt{var}$ (R-L). Both FA1 and FA4a are known to be biased when directional asymmetry or antisymmetry is present [61,62]. Table 2 provides descriptive statistics for the analyzed traits including classical parameters of variation (number of individuals (N), minimal observed value (min), maximal observed value (max), sum of the relative values for (R-L) (sum), mean value for the relative values for (R-L) (mean (R-L)), FA4a: $0.798\sqrt{var}$ (R-L)). Directional asymmetry (DA) corresponds to a pattern of bilateral variation observed in a sample when one side is statistically larger than the other. DA is identified by statistical tests for departures of mean (R-L) from zero [63]. Results of simple t-tests for DA are provided in Table 2. Antisymmetry is a pattern of bilateral variation in a sample when a significant difference exists between sides, but when the larger side varies randomly among individuals [61]. We compare DA, defined as mean (R-L), and FA4a because if DA ≤ FA4a, the predisposition towards one side is less than the average deviation about the mean (R-L) [58]. In those cases, deviations about the mean DA would be due largely to developmental instability (DI). Kurtosis was compared to tables of critical values for platy- and lepto-kurtosis ([58] Table 5, values for Equation (7)). The procedures to test for skewness and corresponding tables for critical values are detailed in Sokal and Rohlf [64]. Results of tests for a departure from normality (skewness, Kurtosis) are provided in Table 2. A sequential Bonferroni procedure was used for post-hoc multiple test correction for all the statistical procedures [61,65].

Statistical procedures were conducted with the Past 2.35 software [66].

We entirely agree with the recently and widely expressed view that the concept of statistical significance is problematic [67]. However, we are dealing in this study with complex concepts (FA, DA, DI, etc.), analyzing anatomical traits that show different sizes, patterns of variation and various degrees and patterns of departure from bilateral symmetry. We have, therefore, followed well-established methodologies to study symmetry [58,61] and used several statistical procedures. Following the suggestion of [67], p-values are reported with sensible precision and do not only refer to significance thresholds. Moreover, we explore the whole meaning of our results, including seemingly "non-significant" or "negative" results. This aspect is also important for the complex study of departures from symmetry.

## 4. Results

First, we tested, for each analyzed trait, whether there were correlations between variations in departures from symmetry and their global size, using the Bonferroni correction for multiple testing. We compared the maximal length of the skull with the values for |R-L| (without any size correction) for all the endocranial traits. The results of the Spearman test of association are non-significant. We have also compared values for |R-L| for all the humeral traits with corresponding trait sizes. The results of the Spearman test of association are non-significant for all the analyzed traits. As a result, all subsequent analyses are conducted on size-corrected data in relation with this first result, but also, and more importantly, in order to allow for a direct comparison of endocranial and humeri data.

We then searched for possible aberrant values in our database of relative measurements. Grubbs' test statistic identified two significant outlier data within our database of size-corrected measurements (7.78, p = 0.035 for the vertical component of the occipital petalia in one individual and 11.41, p = 0.012 for Imax/Imin in another). Those data were excluded from subsequent analyses of asymmetry, explaining why N is only 31 for these two parameters and 32 for all the others (Table 2). We tested the sample for putative sex-related variation and found that the means for males and females do not differ (Hotelling's T-squared tests) for any of the analyzed traits. As a result, all the following analyses are done on the complete pooled sample.

Several parameters were considered to describe the variation for each of the analyzed traits (Table 2) and the distribution for each trait has been explored and is illustrated (Figures 2 and 3).

Shapiro–Wilk tests show that the different traits analyzed all follow a Gaussian distribution, with the exception of the vertical component of the occipital petalia (W = 0.92, p = 0.025).

We searched for indicators of DA. Based on simple t-tests followed by a Bonferroni correction (Table 2), DA is significantly different from zero for the AP component of the frontal petalia (mean = 0.25, t = 2.52, p = 0.034), the AP component of the occipital petalia (mean = −0.66, t = −3.52, p = 0.004), the lateral component of the occipital petalia (mean = 1.87, t = 2.23, p = 0.04), HBIOL (mean = 1.73, t = 6.16, p = 0.00004) and Imax/Imin (mean = −7.56, t = −5.41, p = 0.0003).

Different observations are informative of the patterns of variation around the mean for departures from symmetry (Table 2 and Figure 2). The AP component of the occipital petalia is significantly leptokurtic (2.31, p < 0.05) and significantly skewed (ts = 2.31, p < 0.05). Mean (R-L) is inferior to FA4a for all endocranial traits, as well as for J and %CA. On the contrary, mean value for DA HBIOL (1.73) is superior to the value of FA4a for this trait (1.25). The mean value for DA Imax/Imin (−7.56) is large compared to FA4a (6.33). Among all the analyzed traits, J and Imax/Imin show a large range of variation as expressed by different indicators, including FA4a (Table 2) compared to the other traits. Globally, the variation observed for each of the endocranial traits is lower than that observed on the humeri (Table 2 and Figure 3).

Finally, we considered the possible correlation between endocranial and humeri traits. Among the analyzed sample, 27 individuals (84.4%) follow the classic pattern of RF/LO petalia, as described by Holloway et al. [20], four have a RF/RO pattern and only one has the opposite pattern to the most frequent one, LF/RD. Among these five latter individuals who do not conform to the most common petalial pattern, four share an asymmetry pattern of the upper arm which is only observed in seven of the 32 individuals in the sample, with a right humerus longer that the left one (HBIOL) while all internal parameters related to bone structure show leftward asymmetry (J, %CA, Imax/Imin). A total of 28 individuals have a right humerus longer that the left one (HBIOL) and 27 out of 31 have a value for Imax/Imin which is larger on the left side compared to the right side.

A non-parametric (Spearman's coefficient of rank correlation) test of association was used to test the relationship between the magnitude of the petalias and those of the traits measured on the humeri. After Bonferroni correction, a significant correlation is observed between the antero-posterior and the lateral components of the occipital petalia (rs = −0.-63; p = 0.0018) and between HBIOL and J (rs = −0.57, p = 0.028). No significant correlation is observed between any endocranial trait and any trait quantified on the humeri.

**Table 2.** Descriptive statistics for the analyzed traits on the endocast (namely the AP, VERT and LAT components of the frontal and occipital petalias) and the humeri (HBIOL, J, %CA, Imax/Imin), including classical parameters of variation (number of individuals (N), minimal observed value (min), maximal observed value (max), sum of the relative values for (R-L) (sum), mean value for the relative values for (R-L) (mean (R-L)), FA4a: $0.798\sqrt{var(R-L)}$), indices of asymmetry and tests for directional asymmetry and for departure from normality (skewness, Kurtosis).

| | F.AP | F.VERT | F.LAT | O.AP | O.VERT | O.LAT | HBIOL | J | %CA | Imax/Imin |
|---|---|---|---|---|---|---|---|---|---|---|
| N | 32 | 32 | 32 | 32 | 31 | 32 | 32 | 32 | 31 | 32 |
| Min | −0.9 | −4.9 | −5.6 | −2.6 | −4.8 | −8.6 | −2.1 | −29.3 | −7.4 | −25.7 |
| 25 prcntil | −0.1 | −2.4 | −1.8 | −1.3 | −2.4 | −2.0 | 0.6 | −11.6 | −0.9 | −14.7 |
| Mean (R-L) | 0.2 | −0.1 | −0.1 | −0.7 | −0.6 | 1.9 | 1.7 | −1.3 | 0.4 | −7.6 |
| Median | 0.3 | −0.5 | −0.3 | −0.8 | 0.3 | 2.1 | 1.9 | −0.6 | 0.3 | −6.2 |
| 75 prcntil | 0.6 | 2.2 | 2.0 | −0.1 | 1.2 | 5.3 | 2.8 | 12.8 | 2.2 | −2.6 |
| Max | 1.8 | 5.4 | 4.5 | 2.5 | 2.5 | 12.7 | 5.2 | 21.6 | 5.3 | 12.3 |
| Sum | 7.9 | −2.9 | −1.6 | −21.1 | −17.3 | 59.9 | 55.3 | −41.7 | 13.2 | −241.8 |
| FA4a | 0.4 | 2.1 | 2.1 | 0.8 | 1.7 | 3.7 | 1.3 | 11.2 | 2.9 | 6.3 |
| Std. error | 0.1 | 0.5 | 0.5 | 0.2 | 0.4 | 0.8 | 0.3 | 2.6 | 0.5 | 1.6 |
| Variance | 0.3 | 6.7 | 6.6 | 1.1 | 4.5 | 22.6 | 2.5 | 208.6 | 8.5 | 82.0 |
| Stand. dev | 0.6 | 2.6 | 2.6 | 1.1 | 2.1 | 4.8 | 1.6 | 14.4 | 2.9 | 9.1 |
| ((R-L)/0) t | 2.5 | −0.2 | −0.1 | −3.5 | −1.4 | 2.2 | 6.2 | −0.5 | 0.1 | −5.4 |
| p | $3.4 \times 10^{-2}$ | 0.84 | 0.91 | $4 \times 10^{-3}$ | 0.15 | $3.3 \times 10^{-2}$ | $3.9 \times 10^{-6}$ | 0.61 | 0.93 | $2.9 \times 10^{-5}$ |
| Skewness | 0.4 | 0.3 | 0.1 | 1.0 | −0.6 | −0.1 | −0.1 | −0.2 | −0.5 | −0.1 |
| Ts | 0.9 | 0.7 | 0.2 | 2.3 | −1.5 | −0.2 | −0.3 | −0.6 | −1.2 | −0.2 |
| P | > 0.05 | > 0.05 | > 0.05 | < 0.05 | > 0.05 | > 0.05 | > 0.05 | > 0.05 | > 0.05 | > 0.05 |
| Kurtosis | 1.2 | −0.7 | −0.4 | 1.7 | −0.8 | −0.2 | 0.3 | −0.8 | 0.7 | 0.4 |
| P | > 0.05 | > 0.05 | > 0.05 | < 0.05 | > 0.05 | > 0.05 | > 0.05 | > 0.05 | > 0.05 | > 0.05 |

Frontal (F), occipital (O), anteroposterior (AP), vertical (VERT), lateral (LAT).

## 5. Discussion

Concerning the endocranial petalias, a direct comparison may be done between the results that we have obtained and results obtained with the same methodology in a previous study [2] that comprised a sample of recent European *Homo sapiens*. Mean values for (R-L) and variation as expressed by FA4a are very similar between those two analyzed samples. Moreover, a significant DA is also observed in the two samples for the AP (toward the left) and lateral (toward the right) components of the occipital petalia. The AP frontal petalia is significant on the archaeological sample studied here, while prevalence toward the right side was also observed in our previous sample. The AP occipital petalia is significantly leptokurtic and skewed.

Consequently, the pattern of endocranial asymmetry for the petalias is very similar between the two samples. This is not a surprise, as this pattern is related to the well-known brain torque observed in humans with a forward shift of the right hemisphere relative to the left hemisphere, and lateral expansions of the right frontal lobe and of the left occipital lobe. This pattern concerns traits affected by DA in this study, namely the AP components of the frontal and occipital petalia and the lateral component of the occipital petalia. In addition, the other parameters of the analyzed traits show similar levels of FA in the two samples. This observation is interesting, as it would reflect similar levels of DI for these brain features. Comparison with other samples in the future will help to clarify if petalias may be a good indicator of developmental instability.

In the archaeological sample analyzed here, we observe a significant DA for the length of the humeri, the right one being longer than the left one, associated to a mean (R-L) that is larger than FA4a. The parameter Imax/Imin shows significant larger values on the left side compared to the right side and a large value for mean (R-L) when compared to FA4a. It means that the left humerus was characterized by greater bending strength along the Imax plane than the right upper limb. It is assumed that Imax/Imin can be used as a behavior indicator, however the level of activity that will cause significant changes in shaft shape is still unknown. Moreover, this biomechanical property is very variable within human populations and can express phylogenetic information [68], which is why interpretations of DA Imax/Imin on a small sample size should be drawn with caution. Previous research on the same sample showed that this medieval population was characterized by significant and more pronounced DA of all biomechanical properties (%CA, J and Imax/Imin) than Neandertals and recent Aboriginal Australians [42]. The discrepancy between these results is probably a consequence of analyzing left- and right-handed individuals separately in the last research. A similar methodological approach (assessment of handedness) could not be used in this study due to the smaller sample size. However, a significant DA for the biomechanical length and Imax/Imin can be a consequence of demanding physical tasks that engage one side of the body during the growing years and later on, especially since a bone during development has a great potential of periosteal expansion [68]. High levels of FA4a for the humeral length and Imax/Imin can also suggest that the differences in morphology of the shaft between the left and right humeri are a result of the development phase. Fluctuating asymmetry has been very poorly studied for cross-sectional properties in the past; a comparison between our observations for FA4a here and observations from other research is, therefore, not possible.

As a result, we observe a clear and significant trend of right anterior frontal petalia associated with a left antero-posterior petalia. This is confirmed by the fact that 27 individuals (84.4%) follow the classic pattern of RF/LO petalias as described by Holloway et al. [20]. This pattern has been demonstrated to be highly correlated with right-handedness [19,30,31], whereas left-handers are more often symmetric [69]. Similarly, a trend toward directional asymmetry is observed for some of the traits analyzed on the humeri, a pattern that is commonly associated to right-handedness [70].

However, based on the analyses conducted here, we only observe a correlation between the antero-posterior and the lateral components of the occipital petalia (rs = −0.63; p = 0.0018) and between HBIOL and J (rs = −0.57, p = 0.028). The other parameters do not appear to be significantly correlated. Concerning the occipital petalia, this correlation is due to the fact that the left occipital pole is frequently in a more posterior position, but also in a more mesial position compared to the contralateral side.

Regarding the humeri, the significantly negative correlation between the biomechanical length of the humeri and the standardized J reflects the association between the mechanically relevant distribution of that bone within the cross-section and its length [70].

The indices of robusticity (J) and Imax/Imin show a large variation (Table 2) and we also observe that the variation for each of the traits quantified on the humeri is larger than those observed on the endocast. Imax/Imin also shows DA, whereas J does not, while having the largest value for FA4a or other parameters of variation among the analyzed traits. Diaphyseal bone reveals a high degree of plasticity in response to mechanical loading [71], which is why the normal range of humeral biomechanical properties vary considerably across diverse human populations [72]. The obtained data shows that the level of DA of the upper limb contrasted between individuals of the medieval population. This can be a consequence of analyzing left- and right-handed individuals together. Previous findings on the same group showed that individuals who used, preferentially, the right upper limb exhibit more pronounced DA of the humeral properties than farmers who preferred the left arm [42]. Moreover, the differences in shape of the glenoid cavity suggest that this group was also exposed to unilateral mechanical stimuli [73]. Therefore, the two most variable cross-sectional properties (J and Imax/Imin) may indicate that the medieval population from Ostrów Lednicki was characterized by physically demanding tasks which produced unilateral and bilateral mechanical loading. The endocast reflects the global shape of the brain, whose torque pattern is highly associated to right-handedness. Bilateral variation in the position of the frontal and occipital poles, therefore, shows some evident signs of DA but in the form of a restricted variation—both in terms of direction and proportions—around this clearly oriented pattern. The length and internal structure of the humeri are also affected by manual laterality [1,43,74]. However, those bones are also influenced by large variations in terms of the magnitude of solicitation, explaining the high variation observed here for internal traits. This may result in important phenotypic plasticity, making analyses of FA more complex. We were aware that the biological areas considered here were not ideal models to analyze the departure from symmetry in *Homo sapiens*. DA is, indeed, present because of brain lateralization and lateralized behaviors. However, the combined study of these anatomical elements and of their asymmetries is now proven to shed light on the ways in which the body responds to dependent asymmetrical stimuli across biologically independent anatomical areas. Variations in FA are, in this context, indicators of differences in answers to lateralized factors.

In conclusion, previous independent analyses have shown that manual laterality has some influence both on the torque of the brain, and on the petalias of the endocast, as analyzed here, and on the shape and internal structure of the humeri. However, it is difficult to find a direct correlation between the traits of endocranial asymmetry and humeral bilateral asymmetry when analyzed together. Indeed, they are subject to the influence of a long list of biological, physiological or behavioral factors reflected in the whole set of variations in departures from bilateral symmetry and in patterns of covariation, as observed in this study. It is, therefore, difficult to dissociate on dry bones the influence of such a variety of factors. Moreover, manual laterality and, more generally, behavior, but also morphological variation, cannot be reduced to a simple right-to-left difference. However, we have shown that important but complex information may be extracted from the combined study of the endocast and the arms. Future work on the topic is necessary, exploring departures from symmetry in a detailed manner, as proposed here. These studies will have to include larger samples with a more precise knowledge of all the factors that are involved in the expression of the analyzed traits.

**Author Contributions:** Conceptualization, A.B. and A.M.K.; methodology, A.B. and A.M.K.; software, A.B., L.B.-A. and A.M.K.; validation, A.B., L.B.-A. and A.M.K; formal analysis, A.B. and A.M.K; writing—original draft preparation, A.B., L.B.-A. and A.M.K. All authors have read and agreed to the published version of the manuscript.

**Funding:** This research received funding from the National Science Center in Poland, Grant Number: 2015/19/N/NZ8/00177 and a from the Polish Ministry of Science and Higher Education, Grant Number: 506.511.09.00.

**Acknowledgments:** The authors thank the reviewers for their helpful comments. We thank Anna Wrzesińska from the Museum of the First Piasts at Lednica, for giving the possibility to use skeletal material from Ostrów Lednicki. They also thank Dariusz Kurkiewicz for technical support in scanning the osteological material and Antonio Profico for help in improving the methodology of 3D endocast reconstruction. Anna Maria Kubicka thanks the Polish National Agency for Academic Exchange (NAWA) for the scholarship at the Musée de l'Homme, which allowed to conduct this research.

**Conflicts of Interest:** The authors declare no conflict of interest.

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
