# Peer review of "Variation and Correlations in Departures from Symmetry of Brain Torque, Humeral Morphology and Handedness in an Archaeological Sample of Homo sapiens"

_symmetry, doi:10.3390/sym12030432_

Round 1

Reviewer 1 Report

I am not expert in this field of research but measurement of asymmetry in more precise regions should be necessary. The materials contain both sexes and different age but there are no clear mention about such factor. Analysis of such factors in statistics should be done.

First of all, the results does not provide clear conclusion. Functional asymmetry (handedness) may be correlate with size of particular motor area, so not only overall asymmetry of the brain but also asymmetry of the articular area should be analyzed.

Author Response

Thank you for these helpful comments. We have now included a sentence about gender related variation, that is in fact null.

concerning your second point, we fully agree but unfortunately the brain endocast does not permit to quantify the volume of functional area as identified on the brain. For this reason we have focused on the trait related to hand laterality that is the most studied in the literature and that is visible on endocasts, the petalias. Moreover, this work is a pionner study of related anatomical traits on an archeological sample, future research will contribute to explore in further details those aspects.

Reviewer 2 Report

This manuscript aimed to examine asymmetries present in an archeological sample.  Any asymmetries present would then be used to make assumptions about behavioral asymmetries. 

The literature review and the materials are presented in a clear way.  It was easy to follow and understand the reasons for conducting this study, and from where the materials for this study were derived. 

Are there always imprints left by the brain on the internal surface of the skull?  Or, to conduct a study such as this, only those skulls with an endocast are included?  If it is the case that only skulls with an endocast are included, some mention of why there is sometimes not an endocast would be appreciated.  Is there something similar about skulls that do have an endocast?  Are there unifying reasons that an endocast may be visible, and another unifying reason that an endocast is not present in the other skulls?

The questions above are the only ones that I felt were not addressed in the study.  Clarification at the beginning of the manuscript on this matter would be useful.   

This study provides information relevant to the study of asymmetries, both anatomical and behavioral.    

Author Response

Thank you for your comments. We have added some information about the way brain endocast and the specific analysed traits are observed.